# Position Measurements Using Magnetic Sensors for a Shape Memory Alloy Linear Actuator

**DOI:** 10.3390/s22197460

**Published:** 2022-10-01

**Authors:** Ricardo Cortez Vega, Gabriel Cubas, Marco Antonio Sandoval-Chileño, Luis Ángel Castañeda Briones, Norma Beatriz Lozada-Castillo, Alberto Luviano-Juárez

**Affiliations:** 1Unidad Profesional Interdisciplinaria en Ingeniería y Tecnologías Avanzadas, Instituto Politécnico Nacional, Ciudad de México 07340, Mexico; 2Centro de Investigación y de Estudios Avanzados del Instituto Politécnico Nacional, Ciudad de México 07360, Mexico; 3Unidad Profesional Interdisciplinaria de Energía y Movilidad, Instituto Politécnico Nacional, Ciudad de México 07738, Mexico; 4Centro de Desarrollo e Innovación Tecnológica (CDIT) Vallejo-i, SECTEI, Ciudad de México 01020, Mexico

**Keywords:** shape memory alloy, linear actuator, magnetic sensor, temperature sensor

## Abstract

This article presents the design and implementation of a linear actuator based on NiTi Shape Memory Alloys with temperature and position measurements based on a magnetic sensor array and a set of thermistors. The position instrumentation is contact free to avoid friction perturbations; the position signal conditioning is carried out through the calculation of the response of each magnetic sensor, selecting the closest sensor to ensure accurate results on the full range of movement. Experimental results validate the accuracy of the position sensing with a competitive behaviour.

## 1. Introduction

The recent emergence of new robotic systems, such as soft robotics or soft actuation technologies, has led to the development of alternative sensing and actuation approaches. Among flexible actuators, the Shape Memory Alloys (SMAs) have gained growing interest from both scientific and technological communities due to their large force-to-weight ratio, silent movement generation, and capacity for biological/medical applications (biocompatibility), among others [1,2,3,4]. The aforementioned features have led to a wide variety of practical applications and associated theoretical problems, from the synthesis and construction of the forming materials (on micro and macro scale) to the modelling, variable sensing, and control-oriented of their associated final application [5,6,7].

Concerning the variety of applications, in [8] SMAs were used for deformable wing design to offer a set of different wing shapes to be used in the development of smart morphing wings [9]. Other aerodynamic applications concern the design of morphing structures for the load control in wind generators among other aerospace applications [10,11,12,13]. Other interesting applications are the soft actuation of alternative robotic configurations such as origami-inspired robots [14], bio-inspired robots [15,16], controlled bending antennas [17,18,19], or robots with energy applications [20].

The use of SMAs in robotic applications is also closely related to the design of grippers and actuators [21,22,23,24,25,26] in which the problems of position and force control become quite important. The balance between the correct measurement, modelling, and robust control has motivated several robust control approaches [27,28,29].

Among the reported SMA-based actuators, most are large structures, and they usually consider the position as an interesting variable of measurement. The last fact leads to the use of vision systems and external position sensors (which may imply physical contact with the moving mechanical structure to operate) for control purposes. On the other hand, the temperature information of the SMA is an important variable since it is the physical variable that is directly related to the force/elongation behaviour. Then, a wider variety of control tasks and laws would be possible in case the temperature is available for measurement in an SMA-based actuator. Moreover, a comprehensive thermal characterization of the actuator can be carried out in an on-line fashion if boundary measurements of the temperature in the SMA are available so that potential distributed parameter identification procedures can be performed.

In this article, a portable Shape Memory Alloy linear actuator module with internal position/temperature measurement is reported. The agonist–antagonist configuration is used to carry out both senses of movement. The internal variable sensing is based on an array of magnetic sensors for the position measurement, and thermistors sense the temperature in the extremes of the actuator. The remainder of the article is stated as follows: Section 2 presents the mathematical principles of the SMA-based actuation dynamics, Section 3 deals with the mechanical instrumentation system and the signal conditioning; this also section presents the mechanical design and the general internal interconnections. Section 4 presents the specific position computation procedure based on the magnetic sensor array. The experimental assessment of the system is provided in Section 5, and, finally, some general concluding remarks are given in Section 6.

## 2. SMA Linear Actuator Mathematical Model

The mechanical movement of the SMA linear actuator is generated by two SMA springs placed in an agonist–antagonist scheme to generate the desired movement. This kind of mechanical scheme is shown in Figure 1 where a Mobile Position (MP) modified based on the application of two forces, one agonist and another antagonist, respectively, exists. This kind of system has been implemented to emulate the behaviour of biological systems [30,31,32]. It must be pointed out that this kind of mechanism is highly valuable when some kinds of transmission mechanisms are applied, such as a driven cable [33,34], or when a special class of smart actuator is implemented such as the shape memory alloys [27,35,36], pneumatic actuators [37,38], and electro-chemical actuators [39,40].

### 2.1. Agonist–Antagonist Mechanical Model

The way the agonist–antagonist is used to generate the linear actuator is shown in Figure 2. Notice that a linear shaft allows the generated movement to be freely used to generate motion over other systems such as a robot. Since the linear shaft corresponds to the main interest element from the actuator, its position is defined as *x*, and the dynamic model developed in this work is focused on it.

The dynamics that describe the mechanical movement of the actuator is,
(1)$$m\ddot{x}\left(t\right)=-f_{r}\dot{x}\left(t\right)-k_{1}\left(x-l_{1}\right)+k_{2}\left(x-l_{2}\right)+F_{a}\left(u,t\right)+F_{n}\left(u,t\right)$$
where *m* is the mass of the linear shaft, $f_{r}$ is a term related to the viscous friction, $k_{1}$ is the stiffness from the agonist SMA spring, and $k_{2}$ is the stiffness from the antagonist SMA spring. The terms $l_{1}$ and $l_{2}$ correspond to the values of *x* where the SMA springs, agonist, and antagonist, respectively, are not stretched.

Notice that the defined shape of the SMA (which tends to recover when the temperature is increased) corresponds to a compressed spring such that the workspace of the linear actuator is defined as $x\in [l_{1},L-l_{2}]$, where *L* is the total length of the actuator.

### 2.2. Shape Memory Alloy Spring Model

The forces $F_{a}$ and $F_{n}$ generated by the SMA springs depend on the transition between the martensite and austenite phases on the material. In the martensite phase, the SMA possesses mechanical properties similar to a common metallic alloy such that the only force that influences on the linear movement of the actuator is related to the stiffness. However, when the temperature increases over a threshold value, the SMA tends to recover a predefined shape generating a force during the process. This force is defined using a hybrid function to represent the hysteresis effect without the use of internal variables such as the Dahl model [41], the LuGre model [42] and the Bouc-Wen model [43]. The function $F_{i}$ represents the way $F_{a}$ and $F_{n}$ are calculated. This function depends on the temperature *T* and its first time-derivative $\dot{T}$, and it is defined as [44],
(2)$$F_{i}\left(T,\dot{T}\right)=\left\{\begin{array}{cc} \frac{a_{l}}{1+e^{-b_{l}\left(T-d_{l}\right)}}+c_{l} & \dot{T}<0\\ \frac{a_{u}}{1+e^{-b_{l}\left(T-d_{u}\right)}}+c_{u} & \dot{T}\geq 0 \end{array}\right.$$
where the terms $a_{l},a_{u}$ are related to the maximum force that is generated on the SMA spring, $b_{l},b_{u}$ modify the slope between the minimum and the maximum force generated, $c_{l},c_{u}$ define the minimum force that the SMA spring generates on the hysteresis loop, and $d_{l},d_{u}$ are related to the temperature value in which the change between phases starts. The switching between both cases is given by the sign of the temperature time derivative.

The temperature of the SMA spring is modified by the application of a current *i* between its terminals. This phenomenon can be represented by the following differential equation,
(3)$$\dot{T}=\alpha i^{2}-\beta \left(T-T_{a}\right)$$
where α is the term related to the heat coefficient of the material, and β is related to the heat dissipation from the spring, and it depends not only on the material properties but also on the shape given to the spring.

The dynamics of the temperature on the spring are closely related to the force generated by the SMA spring. A spring with a temperature that has been increased by heating does not stop to generate force when the heating process is stopped; the force related to the martensite–austenite transformation given by (Equation 2) remains until the temperature decreases and the spring completely returns to the martensite phase.

## 3. Instrumentation of the SMA Linear Actuator

The measurement of the position from the linear shaft of the actuator is the main variable that must be acquired. This is performed by the use of an array of magnetic sensors attached to the main body of the actuator. Since the magnitude from a magnetic field decreases by the distance related to the source of the field, the change on the position from the neodymium magnet attached to the linear shaft produces changes in the measurement that could be used to estimate the distance between the magnet and the sensor. The second variable that is instrumented corresponds to the temperature. This one is selected because it is related to the safety of the activation of the actuator since the SMA requires a heating process. An excess in it could damage the materials used for the construction or may produce problems related to the operational temperature limits from the integrated circuits and sensors.

### 3.1. Position Measurement

To estimate the position of the linear shaft, an array of magnetic sensors based on the DRV5053VAQDBZR from *Texas Instruments* was selected to obtain a feasible measurement without adding mechanical elements that increase friction for the movement of the linear shaft. The implementation of an array of sensors is necessary because the magnetic sensors in combination with a dipole neodymium magnet attached to the linear shaft only provide a measurement when there is a narrow distance among them.

The selected sensor possesses an N=±45 mV/mT sensitivity given by the datasheet, and the chosen neodymium magnet possesses an approximate value of B=1.1 Teslas, being the magnitude of the magnetic field B claimed by the manufacturer. However, the magnetic field of the magnet that influences the sensor possesses an abrupt decrease when the distance between them increases. The way the sensor is used to identify the magnetic field magnitude corresponds to a negative sensitivity; this means that if there exists a higher magnetic field, the output voltage decreases such that a minimum value of the voltage is reached when the minimum distance between the sensor and the linear shaft arises. The output voltage *V* of the sensor on the negative sensitivity is given as,
(4)$$V=V_{ref}-BN$$
where $V_{ref}$ is the reference voltage, that is, the maximum value that is reached when the sensor does not detect a magnetic field.

The magnetic field magnitude of a dipole magnet with respect to a sensor in the space could be represented as [45],
(5)$$B=\frac{\mu_{0}}{4\pi }\frac{3(pn)n-p}{r^{3}}$$
where p is the magnetic moment, r=|R| is the distance between the source of the magnetic field and the sensor, R is the 3D position of the sensor considering the sources as origin, and n=R/r is the unit vector that represents the source–sensor direction.

The magnitude and direction of the field vary depending on the position of the sensor to the neodymium magnet. However, since the orientation of the magnet and the sensor is fixed, the relationship between the magnetic field magnitude and the distance is considered a cubic inverse relationship such as $1/r^{3}$. The placement of the magnetic sensors to the linear shaft that possesses the neodymium magnet is given in Figure 3, where it is shown that the distance between them is not directly related to the linear distance of the shaft *c*.

To calculate the distance between the magnet and the sensor over the movement axis of the actuator, denoted by *c*, the following expression is used,
(6)$$c=\sqrt{r^{2}-y^{2}}$$
where *y* is the fixed distance between the magnet and the sensors. Based on this relationship between the position and the magnetic field that is detected by the sensor considering the negative sensitivity, the expected response is shown in Figure 4. The cubic term related to the distance *r* implies that after a distance *c* the magnetic field decreases below the sensitivity level of the sensor, so a single sensor cannot be capable of measuring the full range of movement from the SMA actuator, justifying the implementation of the sensor array.

The estimation of the position is based on an array of magnetic sensors as given in Figure 5. Notice that the selected neodymium magnet provides a magnetic field shown in gray. When a sufficiently large magnitude of this field exists on the range from a sensor, the distance between the magnet and the sensor could be calculated. Since the ranges of the sensors overlap, it is expected that at least two sensors could estimate the distance with respect to the magnet and by using this information to accurately identify the position of the linear shaft. This happens for almost all the positions of the shaft except in the vicinity of the extremes, in which just one sensor detects the magnetic field, and then the information provided by such a sensor is used to estimate the position.

### 3.2. Temperature Measurement

The temperature of the SMA springs must be monitored to avoid several problems such as reaching the Curie temperature that could affect the magnetic field force [46] or to produce deformation over the materials used in the manufacture of the linear actuator. The temperature measurement is performed using **NTCLE100E3682JB0** negative coefficient thermistor from *Vishay* that allows the measurement using a single and compact sensor. Since the thermistors are not linear, it is necessary to implement an equation that converts the resistance value to a temperature value. This is performed using the Steinhart–Hart equation [47] given as,
(7)$$T=A_{1}+A_{2}\mathrm{In}\left(\frac{R}{R_{\mathrm{ref}}}\right)+A_{3}{\mathrm{In}}^{2}\left(\frac{R}{R_{\mathrm{ref}}}\right)+A_{4}{\mathrm{In}}^{3}\left(\frac{R}{R_{\mathrm{ref}}}\right)$$
where *R* is the thermistor reference, $R_{\mathrm{ref}}$ is the resistance value at a reference temperature being at 25 °C for the employed thermistor with a value of $R_{\mathrm{ref}}$ = 6800 Ω. The constant terms $A_{1}$, $A_{2}$, $A_{3}$, and $A_{4}$ are related to the material properties, and they are given by the manufacturer on the datasheet.

To acquire the temperature signal, an analog-to-digital converter (ADC) from an embedded controller is used to process the voltage signal of the thermistor in a voltage divider configuration as shown in Figure 6.

Such voltage related to the output is given by,
(8)$$V_{out}=V_{cc}\frac{R_{th}}{R_{d}+R_{th}}$$

### 3.3. Mechanical Design and Integration

The mechanical design of the SMA linear actuator is shown in Figure 7. The way the agonist–antagonist scheme is implemented based on two SMA springs that move a linear shaft is exemplified. The placement of the temperature sensors is on the edges of the actuator. Since the spring is fixed to these elements, the contact between the thermistor and the SMA is ensured.

The placement of the sensor array on the SMA linear actuator is given in Figure 8, where each sensor has its section to be attached such that an equidistant array is ensured.

## 4. Position Estimation Based on a Magnetic Sensor Array

Since the measurement of the position of the actuator not only depends on a single sensor but on a sensor array placed on a linear configuration (in this case consisting of five sensors), it is necessary to implement an algorithm that, from the individual sensor measurements, provides the position estimation of the actuator.

To generate the desired algorithm, the positions of the linear shaft where the minimum value of the output for each sensor is measured must be identified. For the SMA linear actuator, the maximum field position designated as γ and its corresponding output voltage are given in Table 1 as well as the reference voltage $V_{ref}$ for each sensor, respectively.

The range of measurement from the distance of the linear shaft for each sensor is given in Table 2. Notice that an overlap exists between their ranges that allows avoiding non-measurable positions. The given configuration of the sensor array results in a mean range of sensors 2, 3, and 4 from 29.06 mm. For the case of sensors 1 and 5, their range is limited because they are located in the extreme positions of the array.

Since the overlap between the positions (shown in Figure 9) allows us to consider that sensor *n* still gives information from the position until at least the medium of the range from sensor n+1, then it could be identified if the measured position corresponds to the right or the left from the minimum value given by the sensor based on which sensors are being measured.

The way used to estimate the position of the linear shaft is defined in Algorithm 1. The proposed solution requires that the placement of the magnetic sensors fulfill the previously defined properties. The algorithm first checks what sensors are actually measuring the magnetic field. Then, it generates a set of flags $F_{i}\;\forall \;i=1,\cdots ,5$ that represent the status of the array. If the sensor is measuring a magnetic field, then $F_{i}=1$ on the other case when no magnetic field is generating a change on the sensor output then $F_{i}=0$. To consider that the sensor is actually measuring the magnetic field, the condition $V_{i}<0.9V_{ref,i}$ is proposed such that the voltage decreases at least 10% of the reference voltage. Then, the range of the solutions to be solved is obtained based on the combination of the active flags $F_{a}$ and $F_{b}$ that are active to interpolate the position based on the following equation,
(9)$$\tilde{x}=\frac{\left(\gamma_{a}+\sqrt[{3}]{|V_{\min,a}-V_{a}|}G\right)+\left(\gamma_{b}-\sqrt[{3}]{|V_{\min,b}-V_{b}|}G\right)}{2}$$
where *G* is a gain term related to the magnetic field generated by the neodymium magnet.
**Algorithm 1:** Position estimation the shaft from a linear actuator using a magnetic sensor array
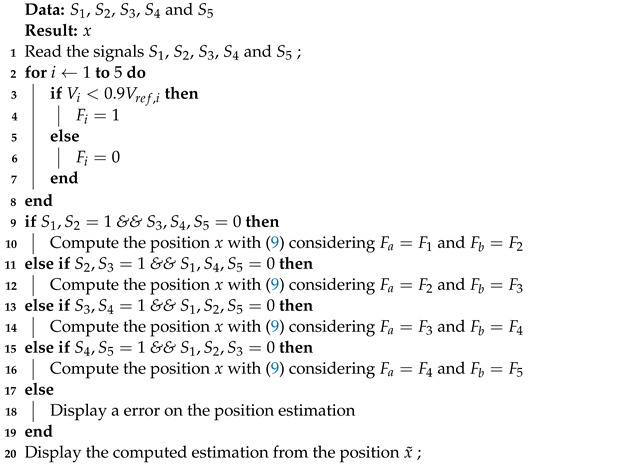


## 5. Results

### 5.1. Experimental Setup

The acquisition of the data related to the sensors was performed with the use of an embedded processor **STM32F4DISCOVERY** which was programmed on Matlab/Simulink environment with the Waijung blockset package using a 12-bits ADC module. A USB/Serial communication was configured between the STM32F4DISCOVERY to a Simulink session with the Runge-Kutta integration method and a sampling time of 0.005 s. The SMA springs on the linear actuator were preset using the procedure given in [44] using a 375 μm of diameter wire from the Dynalloy company. The parameters of the SMA springs are given in Table 3. Note that the parameters $d_{l}$ and $d_{u}$ are related to the transformation temperature.

To condition the data acquired with the ADC from the acquisition board, an FIR discrete filter on a direct form was used, defined as,
(10)$$S_{i,f}\left(n\right)=h\left[0\right]S_{i}\left(n\right)+h\left[1\right]S_{i}\left(n-1\right)+\cdots +h\left[N\right]S_{i}\left(n-N\right)$$
where $S_{i,f}$ is the filtered signal from $S_{i}$, and *h* is a vector that contains all the coefficients from the filter, that is $h=h_{0},h_{1},\cdots ,h_{N}$. The frequency domain filter is represented by the following expression,
(11)$$F\left(z\right)=\frac{h_{0}+h_{1}z^{-1}+h_{2}z^{-2}+\cdots +h_{N}z^{-N}}{1}$$

The coefficients from the filter were selected using the **Filter Design and Analysis Tool** from Matlab, which considers a frequency pass from 10 Hz, a stop frequency of 15 Hz, a sampling frequency of 200 Hz, an amplitude from the pass-band of 1dB, and amplitude of the rejected-band of −80 dB. The resulting filter provides a total from N=101 and with a response shown in Figure 10, which shows that the resulting filter provides a cut-off frequency that attenuates the high-frequency components from the signal necessary to eliminate the high-frequency noise components during the instrumentation process.

The linear actuator presented in this work is designed to be activated with the implementation of an electric current *i* that produces an increase of the temperature on the spring such that the force generated by the change between phases provides the motion. The applied current to the SMA springs is 1.6 A, with a voltage of 4 V provided by a voltage source of 20 W. To activate the SMA spring that generates the movement of the linear shaft, an open-loop scheme was implemented to supply the current to the spring to increase the temperature and produce the transition between the martensite and austenite phases.

### 5.2. Mechanical Structure

The mechanical structure of the linear actuator was manufactured using PLA with an M200 Zortrax printer for the main body of the actuator. However, the elements that are in contact with the SMA springs (the PLA is not a feasible option for these parts since the temperature could produce deformation on them) such as the linear shaft and the covers were manufactured using a Siraya Tech high-temperature-resistant resin and a Phrozen 4K resin printer, to avoid high-temperature deformations from the Nitinol when reaching the austenite phase.

The resulting actuator is given in Figure 11, where the sensors are placed on one side of the actuator as established in the mechanical design. Thus, the linear shaft moves to the sensor array freely.

The linear shaft that is placed in the middle of the actuator is shown in Figure 12. This lateral view allows us to see the placement of the neodymium magnet that activates the sensor array. The dimensions of the linear shaft ensure that the magnet position is at the same height as the sensor array, such that the distance between them is only increased by the change of position over the movement axis of the linear shaft.

The actuation with the agonist–antagonist method is given in Figure 13. It shows the position of the SMA that allows for generating the opposition force between them. The linear shaft moves linearly between the compressed position of SMA spring 1 and the compressed position of SMA spring 2. The placement of the thermistor for each SMA spring allows physical contact between the sensor and an SMA spring terminal. Since the SMA is a material of metallic nature, its temperature is assumed to be homogeneous.

The manufactured linear actuator based on SMA springs possesses the following characteristics:Operation range of linear displacement from 0 mm to 59.9 mm.Operation current from [1 A–3 A] in order to avoid damage on the mechanical structure.Force value of 2.89 N for each SMA spring.Total weight of 35.6 g including the mechanical structure, SMA springs and sensors.

### 5.3. Instrumented Variables

The measured data from the position obtained from the actuator is depicted in Figure 14. Note that during the first 100 samples, the signal was highly distorted; this happens because the implementation of the filter structure (Equation 11) implies at least to have the first $S_{i}\left(n-N\right)$ samples for a complete processing, leading to transient behaviour with incomplete information.

The experimental data show the process of a moving shaft measurement from positions 0 mm to 59.9 mm on a time interval from 2.5 s to 6.1 s. From the reported results, the placement of the sensors allows implementing the proposed algorithm to estimate the position of the actuator.

The measurement of the temperature using the thermistor, the estimation of the thermistor resistance based on (Equation 8), and the approximation of the temperature based on the change in the resistance given by (Equation 7) are shown in Figure 15. Notice that during the initial samples, the signal provides an error caused by the implemented digital filter. This means that the acquisition stage demands a minimal waiting time of 0.5 s to provide a feasible estimation of the variables to be monitored.

### 5.4. Measurement Position Estimation Based on a Magnetic Sensor Array

The results of the implementation of the method used for the position estimation using the magnetic sensor array are depicted in Figure 16. These positions were calculated using a parameter G=20. It is shown that only once the signals from the magnetic sensors provide feasible measurements (when *N* samples have used the estimation of the position) is a feasible solution obtained.

The estimated output shows that around 3.2 and 4 s the estimated actuator position does not change; this phenomenon is related to a light stall of the linear shaft that also could be noticed in Figure 14. The noise on the estimation of the position is related to the noise in the signal used by the algorithm. It is expected that once these signals are improved, the output of the algorithm improves.

A comparison between the position estimated with the algorithm and a direct measurement obtained with a linear movement potentiometer attached to the shaft of the actuator is shown on Figure 17. The figure illustrates the differences between the estimated position and the linear measurement. This is more explicit on Figure 18, where the error between the estimation and the measurement is presented. From the last Figures, it is clear that the maximum calculated error is around the initial time, and it is related to the filtering of the signals used to evaluate in the algorithm. However, at around 7 s, an error on the estimation of the position of nearly 5 mm is obtained. This is related to the change between the sensors that are detected as active. This implies some future corrections in certain design parameters such as the threshold voltage.

## 6. Conclusions

Based on the given results, the following conclusions can be summarized:The measurement of the position of a linear actuator based on SMA using a magnetic sensor array has been proven to be a feasible option for this kind of system.The placement of the magnetic sensors must be carefully selected to allow the overlap between their measurement range such that the proposed methodology is feasible to estimate the position of the linear shaft.The implementation of the temperature measurement based on thermistors shows an accurate measurement of the status of the SMA springs, and it can be used as part of temperature control, a temperature turn-off indicator for security purposes, or other signal processing/identification approaches.The implementation of discrete filters in direct form provides a valuable alternative to processing the signal. However, the required time that must be waited until the number of samples taken surpasses the number of coefficients on the filter implies that other types of filters must be considered to overcome this problem.The identification from the threshold voltage related used to determine if the neodymium magnet is on range from the sensor must be carefully performed in order to avoid errors that may reduce the accuracy of the estimation.The linear actuator and the manner that the sensor array provides measurements could be improved by using magnetic sensors with higher accuracy and with the replacement of the neodymium magnet by another with a larger magnetic moment value.

## 7. Future Work

The development of the actuator and its corresponding instrumentation for the position measurement based on a linear array of magnetic sensors is just a first stage for the solution of some related problems to this kind of actuators. Two of the most promising topics to be further studied are the following:Stall on the linear movement by torsion of the SMA spring during the heating: During the shape transformation produced by the marternsite–austenite phase change, the spring could produce torsional movements that may generate a stall in the actuator. Since this work is related to the position estimation, this kind of effects is not part of our scope, but this remains an open issue that could be studied since a mechanical approximation related to the configuration of the springs, the manner that they can be placed in the actuator, and the dynamics that may describe the torsional effects could be included as a disturbance term related to the position of the actuator. Another element of the disturbance function that can be further analysed is the temperature related to the geometry of the spring [48,49].Temperature estimation of the SMA spring: In this work, the temperature was just measured on a specific place of the spring, and the obtained information was just acquired in order to avoid damages in the mechanical structure by the overheat during the activation of the actuator. However, the temperature on the SMA spring could be described by a model of distributed parameters where the temperature over the spring is not assumed to be uniform [50,51]. The estimation of parameters from this class of systems requires a highly complex computation effort [52,53]. One of the main issues for the particular case of the developed actuator is that sensor-stage-embedded classical approximations measuring the temperature, such as thermal cameras, must be discarded. Other approximations must be developed.Temperature hysteresis: The time transfer between temperature and their measurement in a feedback can produce a temperature hysteresis that worsens the precision in the cooling path [22,54,55,56]. To solve this open issue, the limitations about the sensors to be included in the embedded linear actuator must be carefully considered.

## Figures and Tables

**Figure 1 sensors-22-07460-f001:**
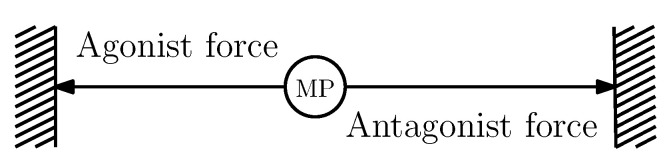
Agonist–antagonist mechanical scheme.

**Figure 2 sensors-22-07460-f002:**
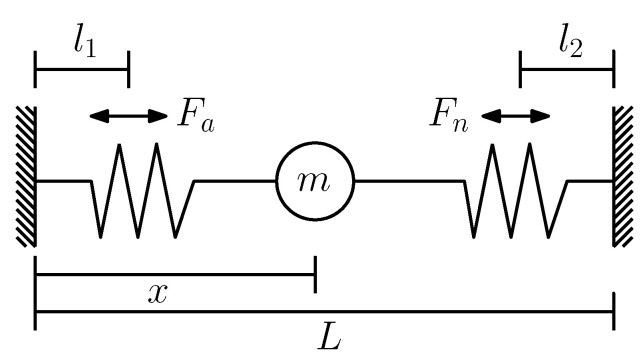
Agonist–antagonist system based on SMA springs.

**Figure 3 sensors-22-07460-f003:**
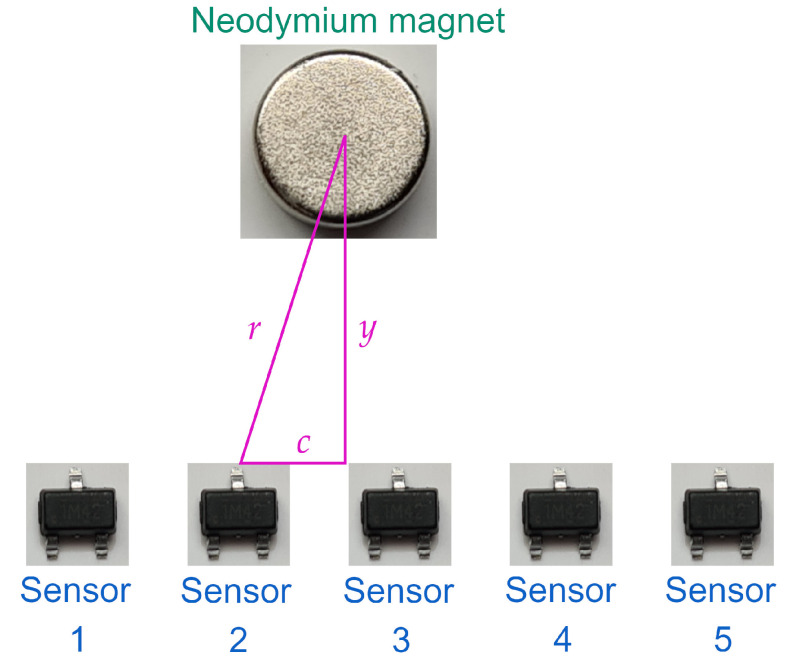
Magnetic sensor array placement. Each sensor is enumerated from $S_{1}$ to $S_{5}$; *r* is the distance between the magnet *M* and the considered sensor; *y* is a fixed distance that exists between the sensors and the linear shaft; and *c* is the distance along the axis of the linear shaft between the magnet and the sensor.

**Figure 4 sensors-22-07460-f004:**
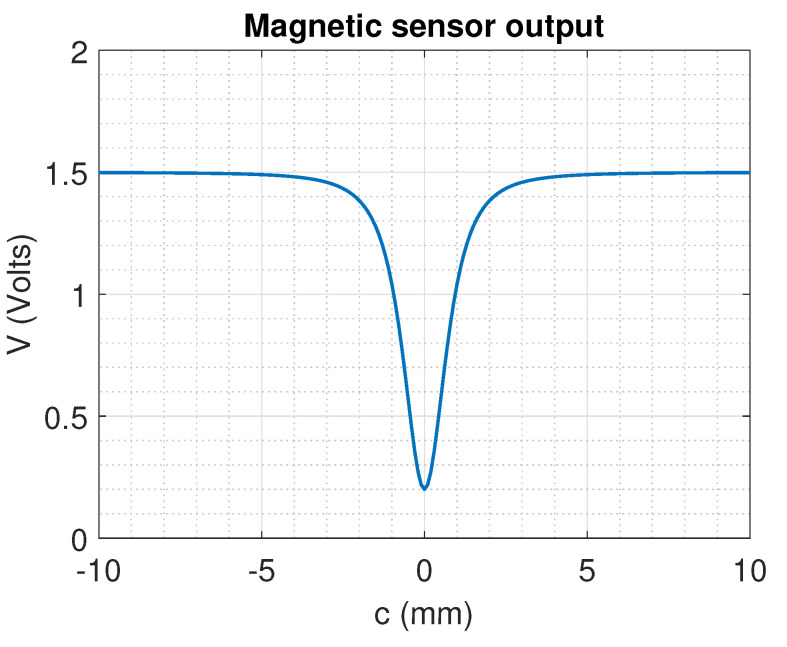
DRV5053VAQDBZR voltage output expected for each sensor based on the distance over the linear shaft movement axis.

**Figure 5 sensors-22-07460-f005:**
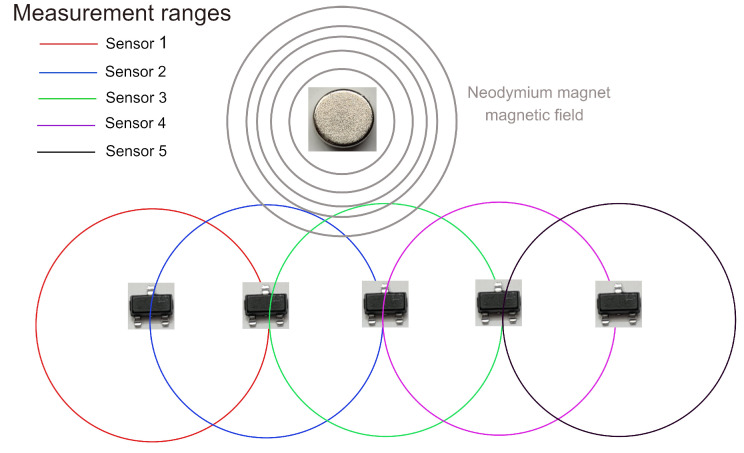
Array configuration of the magnetic sensors and their sensing ranges.

**Figure 6 sensors-22-07460-f006:**
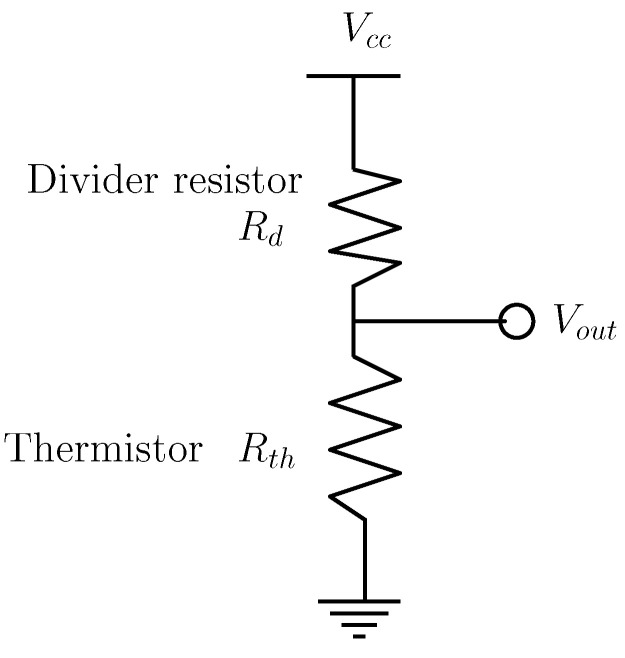
Voltage divider circuit to acquire the temperature data from the thermistor.

**Figure 7 sensors-22-07460-f007:**
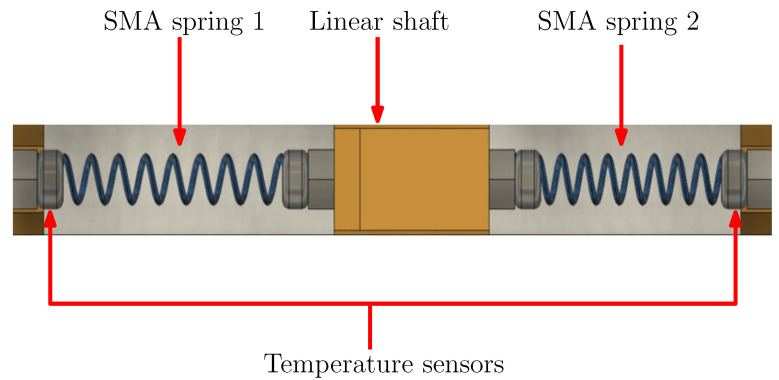
Upper view of the SMA linear actuator design.

**Figure 8 sensors-22-07460-f008:**
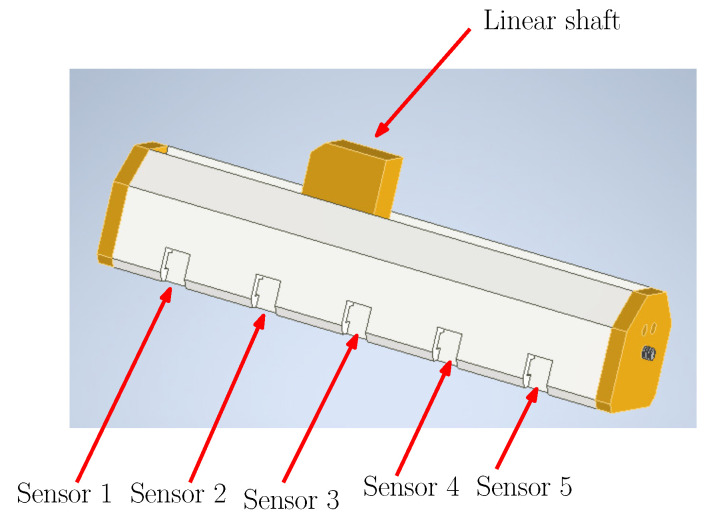
Lateral view from the SMA linear actuator design.

**Figure 9 sensors-22-07460-f009:**
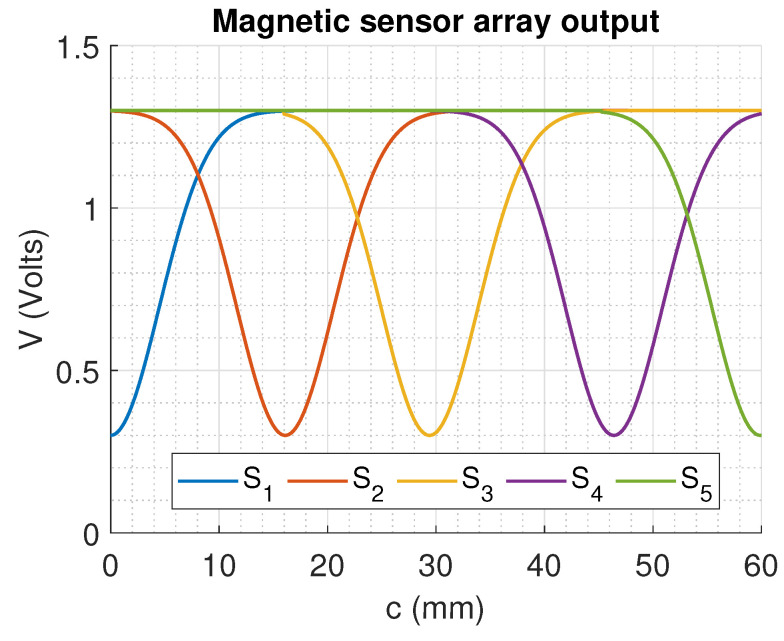
Ideal output from the magnetic sensor array, considering that all the actuators have been placed such that the sensing range between them overlaps.

**Figure 10 sensors-22-07460-f010:**
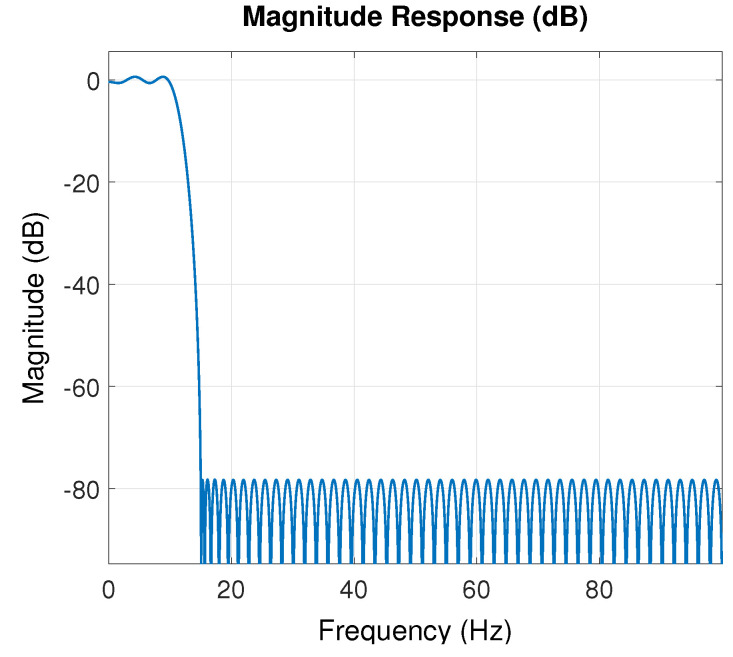
Frequency response from the designed filter using the **Filter Design and Analysis Tool** from Matlab.

**Figure 11 sensors-22-07460-f011:**
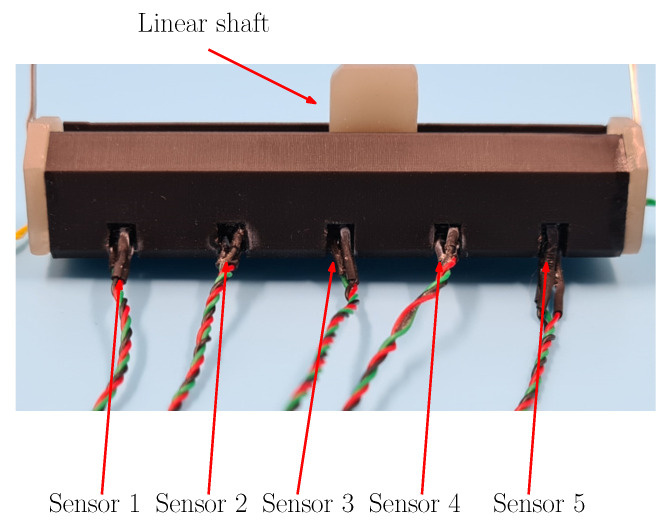
SMA linear actuator and the sensor array from magnetic actuators.

**Figure 12 sensors-22-07460-f012:**
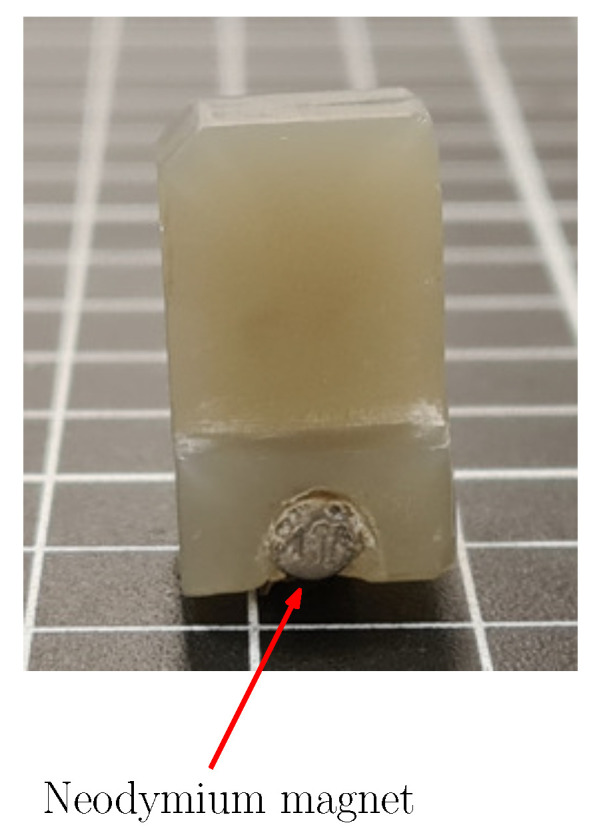
Lateral view from the linear shaft. The placement of the neodymium magnet allows it to be on the same height as the magnetic sensor array.

**Figure 13 sensors-22-07460-f013:**
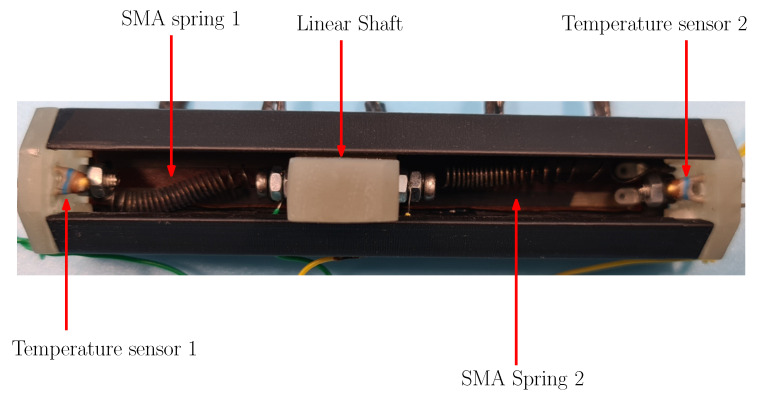
Upper view of the SMA linear actuator structure.

**Figure 14 sensors-22-07460-f014:**
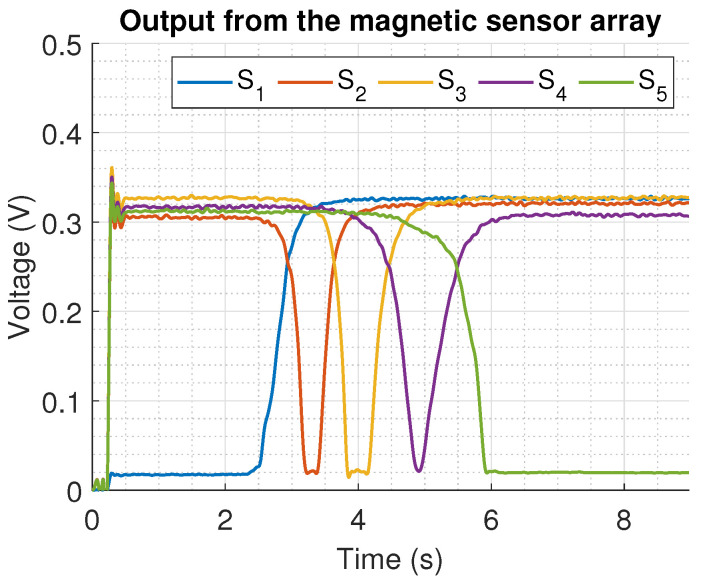
Measured output from the array sensor during the movement of the linear shaft from the position 0 mm to 59.9 mm.

**Figure 15 sensors-22-07460-f015:**
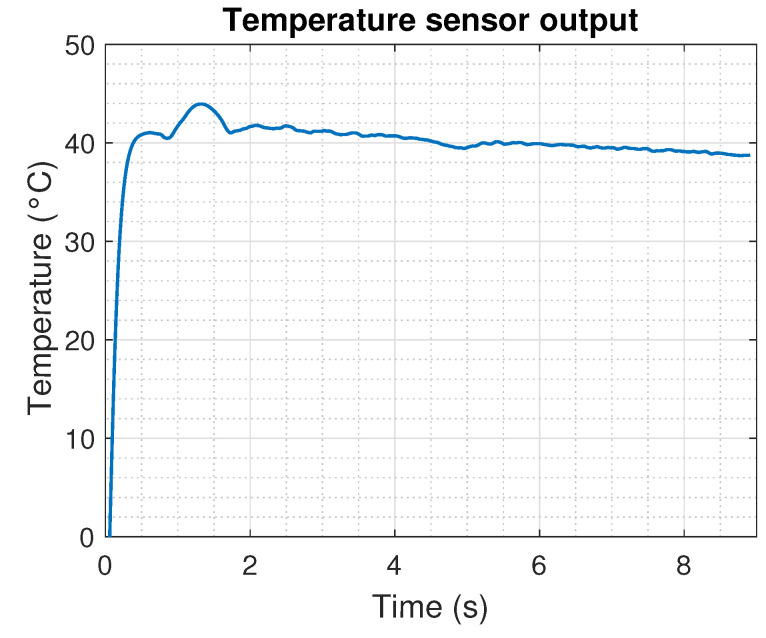
Temperature measurement from the SMA using the thermistor.

**Figure 16 sensors-22-07460-f016:**
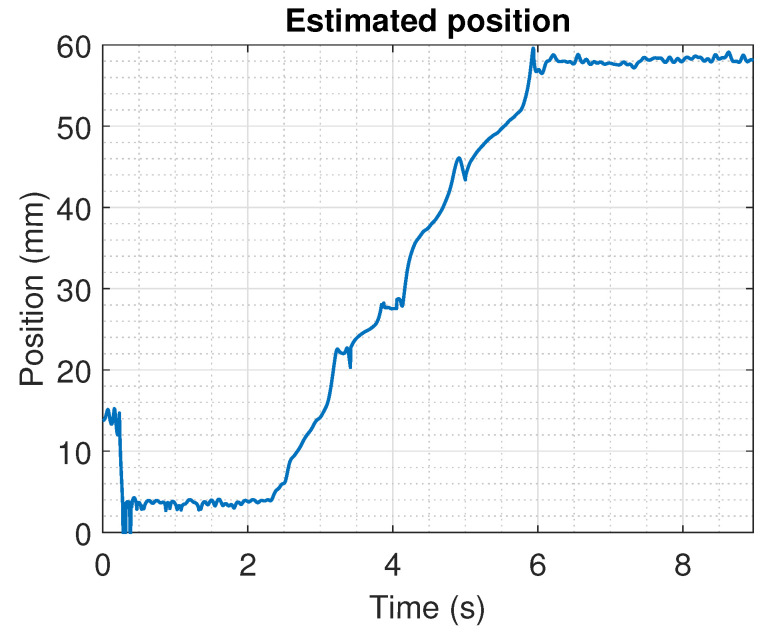
Estimated position using the data obtained from the sensor array.

**Figure 17 sensors-22-07460-f017:**
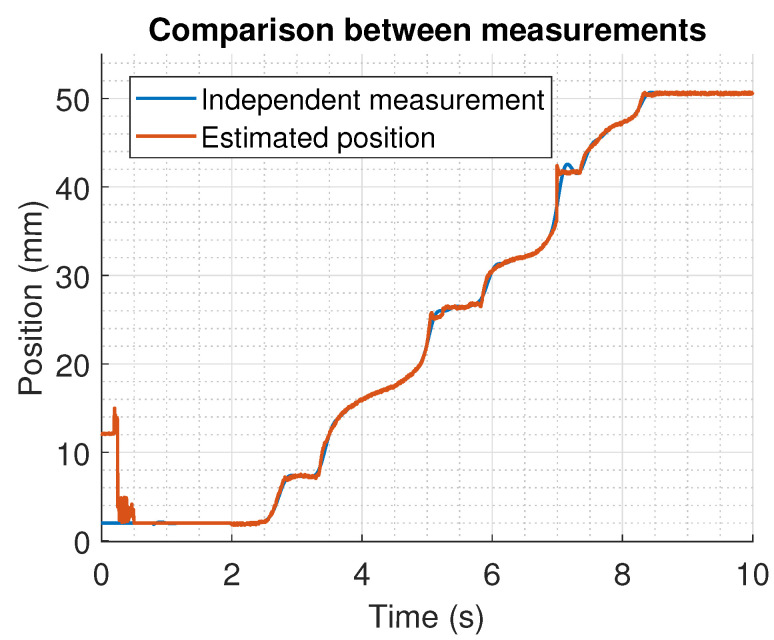
Estimated position with the proposer algorithm and a direct measurement obtained with a linear movement potentiometer.

**Figure 18 sensors-22-07460-f018:**
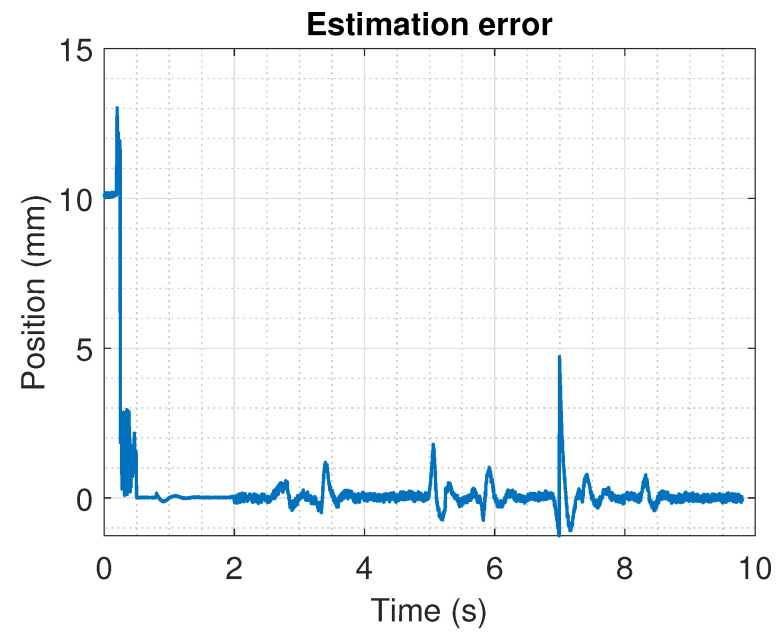
Comparison error between the estimated position and a direct measurement obtained with a linear movement potentiometer.

**Table 1 sensors-22-07460-t001:** Position of the linear shaft where each sensor reaches its minimum value from the output and its reference voltage.

Sensor	γ (mm)	$V_{\min}$ (V)	$V_{\mathrm{ref}}$ (V)
Sensor 1	0	0.017	0.325
Sensor 2	16.1	0.016	0.315
Sensor 3	29.4	0.017	0.325
Sensor 4	46.4	0.018	0.305
Sensor 5	59.9	0.019	0.310

**Table 2 sensors-22-07460-t002:** Position of the linear shaft where each sensor reaches its minimum value from the output and its reference voltage.

Sensor	Minimum Position (mm)	Maximum Position (mm)	Range (mm)
Sensor 1	0	17.2	17.2
Sensor 2	3.6	31.2	27.6
Sensor 3	16.0	47.8	31.8
Sensor 4	28.9	56.7	27.8
Sensor 5	45.3	59.9	14.6

**Table 3 sensors-22-07460-t003:** Parameters of the SMA springs.

Parameter	Value
$a_{u}$	2.89
$b_{u}$	7.69
$c_{u}$	0
$d_{u}$	41.66
$a_{l}$	2.89
$b_{l}$	3.42
$c_{l}$	0
$d_{l}$	38.74
α	30.57
β	5.61
$T_{a}$	25.6

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
