# Peer review of "Position Measurements Using Magnetic Sensors for a Shape Memory Alloy Linear Actuator"

_sensors, 2022, doi:10.3390/s22197460_

Round 1
Reviewer 1 Report
1. The processing of the sensors has been described with very few figures, which make it hard to understand the detail of the design, please add related information in 3.1 and 3.2.
2. Same as other SMA driver, the spring is always slack without heating as show in Fig 12, the control strategy is a simple close-loop control based on position feedback, so what is the effect of the equations (2)-(3)?
3. For the temperature sensor, because the article is without the detail structure of the sensor, it is hard to determine the situation, but normally the time of heat transfer make the feedback value is hysteresis. Which method is used to deal with the situation, and is it related with the frequency.?
4. The format and expression of all the Figs should be consistent. In general, the expression needs to be optimized, the text needs to be further optimized and concise, and the detailed information needs to be clearly marked in the form of charts.
5. From the result of Fig 15, the control strategy is better to be improved.
6. Some basic parameters should be introduced such as transforming temperature.
7. In the application of the SMA based actuator, beside all the effects have been mentioned, the fatigue property and control strategy are also related to the design. Some information have been mentioned in these articles:”Performance analyses of antagonistic shape memory alloy actuators based on recovered strain”,”Multifeedback Control of a Shape Memory Alloy Actuator and a Trial Application ”, and other more.
Reviewer 2 Report
The manuscript presents the position estimation based on magnetic sensor array output to detect a NdFeB magnet position. The study and presentation looks good overall. I have 1 major 1 minor comment:
Major Comment:
Fig. 15, the estimated position data, has to be presented together wit the ground truth data. Without ground truth data, it is hard to analyze if this mechanism is precise or not. The error with ground data should be presented visually and in text.
Minor Comment:
Minor: Line 91 - 92, the end of the sentence is not clear, an English check is required.
Reviewer 3 Report
The paper presents a study on the position and temperature measurements of a shape memory alloy linear actuator. The paper is written and organised well. However, its technical competency needs improvement.
Comments:
1. The literature survey is incomplete. Similar or antecedent models of the actuator should be presented with their demerits and how this paper attempts to address them.
2. Only the design and fabrication of the actuator were discussed. The operation analysis and working capabilities of the actuator need to be addressed.
3. “The experimental data shows the process of a moving shaft measurement from positions 0mm to 59.9mm on a time interval from 2.5s to 6.1s.” What was the method used for the actuation of springs? In the case of electrical actuation, what was the current and voltage used? The dimension and phase transformation temperature of the springs should be given.
Reviewer 4 Report
In this manuscript, a linear actuator based on NiTi Shape Memory Alloys with temperature and position measurements is proposed and analyzed. The implementation of the temperature measurement based on thermistors shows an accurate measurement of the status of the SMA springs. There are a few flaws that need to be addressed in detail:
1. The introduction about the potential applications of the proposed design should be further detailed in the introduction section to elaborate the advantages of such design.
2. As presented in Figure 12, the SMA spring is not linear compressed during the activation of the actuator. Could this distortion affect the accuracy of the modeling and the corresponding results? In addition, the temperature sensor is implemented at the bottom of the SMA spring. How to guarantee the unity of the temperature distribution on the whole SMA spring and what is the effect of such deviation? I think those should be further discussed.
3. Some text seems missing at line 148.
4. Typos and grammatical errors should be revised through the whole manuscript, such as Line 76-77: the SMA tends to recover a predefined shape generating and a force during this process.
5. The format of presented figures (Figure 4, 8, 9, 13-15) should be unified, including the text, boundaries, line color etc.
Round 2
Reviewer 1 Report
No more comments
Reviewer 3 Report
The paper can be accepted.
Reviewer 4 Report
The author answered all my questions very well and I think this paper is now acceptable.